# Leveraging Cognitive Features for Sentiment Analysis

## Abstract

Sentiments expressed in user-generated short text and sentences are nuanced by subtleties at lexical, syntactic, semantic and pragmatic levels. To address this, we propose to augment traditional features used for sentiment analysis and sarcasm detection, with cognitive features derived from the eye-movement patterns of readers. Statistical classification using our enhanced feature set improves the performance (F-score) of polarity detection by a maximum of 3.7% and 9.3% on two datasets, over the systems that use only traditional features. We perform feature significance analysis, and experiment on a held-out dataset, showing that cognitive features indeed empower sentiment analyzers to handle complex constructs.

## 1 Introduction

This paper addresses the task of Sentiment Analysis (SA) - automatic detection of the sentiment polarity as positive versus negative - of user-generated short texts and sentences. Several sentiment analyzers exist in literature today (Liu and Zhang, 2012). Recent works, such as Kouloumpis et al. (2011), Agarwal et al. (2011) and Barbosa and Feng (2010), attempt to conduct such analyses on user-generated content. Sentiment analysis remains a hard problem, due to the challenges it poses at the various levels, as summarized below.

### 1.1 Lexical Challenges

Sentiment analyzers face the following three challenges at the lexical level: (1) **Data Sparsity**, *i.e.,* handling the presence of unseen words/phrases. (e.g., *The movie is messy, uncouth, incomprehensible, vicious and absurd*) (2) **Lexical Ambiguity**, *e.g.,* finding appropriate senses of a word given the context (e.g., *His face fell when he was dropped from the team* vs *The boy fell from the bicycle*, where the verb "fell" has to be disambiguated) (3) **Domain Dependency**, tackling words that change polarity across domains. (*e.g.,* the word *unpredictable* being positive in case of *unpredictable movie* in movie domain and negative in case of *unpredictable steering* in car domain). Several methods have been proposed to address the different lexical level difficulties by - (a) using WordNet synsets and word cluster information to tackle lexical ambiguity and data sparsity (Akkaya et al., 2009; Balamurali et al., 2011; Go et al., 2009; Maas et al., 2011; Popat et al., 2013; Saif et al., 2012) and (b) mining domain dependent words (Sharma and Bhattacharyya, 2013; Wiebe and Mihalcea, 2006).

### 1.2 Syntactic Challenges

Difficulty at the syntax level arises when the given text follows a complex phrasal structure and, *phrase attachments* are expected to be resolved before performing SA. For instance, the sentence *A somewhat crudely constructed but gripping, questing look at a person so racked with self-loathing, he becomes an enemy to his own race.* requires processing at the syntactic level, before analyzing the sentiment. Approaches leveraging syntactic properties of text include generating dependency based rules for SA (Poria et al., 2014) and leveraging local dependency (Li et al., 2010).

### 1.3 Semantic and Pragmatic Challenges

This corresponds to the difficulties arising in the higher layers of NLP, *i.e.,* semantic and pragmatic layers. Challenges in these layers include handling: (a) Sentiment expressed implicitly (*e.g., Guy gets girl, guy loses girl, audience falls asleep.*) (b) Presence of sarcasm and other

forms of irony (*e.g., This is the kind of movie you go because the theater has air-conditioning.*) and (c) Thwarted expectations (*e.g., The acting is fine. Action sequences are top-notch. Still, I consider it as a below average movie due to its poor story-line.*).

Such challenges are extremely hard to tackle with traditional NLP tools, as these need both linguistic and pragmatic knowledge. Most attempts towards handling *thwarting* (Ramteke et al., 2013) and *sarcasm and irony* (Carvalho et al., 2009; Riloff et al., 2013; Liebrecht et al., 2013; Maynard and Greenwood, 2014; Barbieri et al., 2014; Joshi et al., 2015), rely on distant supervision based techniques (*e.g.,* leveraging hashtags) and/or stylistic/pragmatic features (emoticons, laughter expressions such as "lol" *etc*). Addressing difficulties for linguistically well-formed texts, in absence of explicit cues (like emoticons), proves to be difficult using textual/stylistic features alone.

### 1.4 Introducing Cognitive Features

We empower our systems by augmenting cognitive features along with traditional linguistic features used for general sentiment analysis, thwarting and sarcasm detection. Cognitive features are derived from the eye-movement patterns of human annotators recorded while they annotate short-text with sentiment labels. Our hypothesis is that cognitive processes in the brain are related to eye-movement activities (Parasuraman and Rizzo, 2006). Hence, considering readers' eye-movement patterns while they read sentiment bearing texts may help tackle linguistic nuances better. We perform statistical classification using various classifiers and different feature combinations. With our augmented feature-set, we observe a significant improvement of accuracy across all classifiers for two different datasets. Experiments on a carefully curated held-out dataset indicate a significant improvement in sentiment polarity detection over the state of the art, specifically text with complex constructs like irony and sarcasm. Through feature significance analysis, we show that cognitive features indeed empower sentiment analyzers to handle complex constructs like irony and sarcasm. Our approach is the first of its kind to the best of our knowledge.

The rest of the paper is organized as follows. Section 2 presents a summary of past work done in traditional SA and SA from a psycholinguistic point of view. Section 3 describes the available datasets we have taken for our analysis. Section 4 presents an our features that comprise both traditional textual features, used for sentiment analysis and cognitive features derived from annotators' eye-movement patterns. In section 5, we discuss the results for various sentiment classification techniques under different combinations of textual and cognitive features, showing the effectiveness of cognitive features. In section 7, we discuss on the feasibility of our approach before concluding the paper in section 8.

### 2 Related Work

Sentiment classification has been a long standing NLP problem with both supervised (Pang et al., 2002; Benamara et al., 2007; Martineau and Finin, 2009) and unsupervised (Mei et al., 2007; Lin and He, 2009) machine learning based approaches existing for the task.

Supervised approaches are popular because of their superior classification accuracy (Mullen and Collier, 2004; Pang and Lee, 2008) and in such approaches, feature engineering plays an important role. Apart from the commonly used bag-of-words features based on unigrams, bigrams etc. (Dave et al., 2003; Ng et al., 2006), syntactic properties (Martineau and Finin, 2009; Nakagawa et al., 2010), semantic properties (Balamurali et al., 2011) and effect of negators (Ikeda et al., 2008) are also used as features for the task of sentiment classification. The fact that sentiment expression may be complex to be handled by traditional features is evident from a study of comparative sentences by Ganapathibhotla and Liu (2008). This, however has not been addressed by feature based approaches.

Eye-tracking technology has been used recently for sentiment analysis and annotation related research (apart from the huge amount of work in psycholinguistics that we find hard to enlist here due to space limitations). Joshi et al. (2014) develop a method to measure the sentiment annotation complexity using cognitive evidence from eye-tracking. Mishra et al. (2014) study sentiment detection, and subjectivity extraction through anticipation and homing, with the use of eye tracking. Regarding other NLP tasks, Joshi et al. (2013) proposed a studied the cognitive aspects if Word Sense Disambiguation (WSD) through eye-

|     | NB |     |     | SVM |     |     | RB |     |     |
| --- | --- | --- | --- | --- | --- | --- | --- | --- | --- |
|     | **P** | **R** | **F** | **P** | **R** | **F** | **P** | **R** | **F** |
| D1 | 66.15 | 66 | **66.15** | 64.5 | 65.3 | **64.9** | 56.8 | 60.9 | **53.5** |
| D2 | 74.5 | 74.2 | **74.3** | 77.1 | 76.5 | **76.8** | 75.9 | 53.9 | **63.02** |

Table 1: Classification results for different SA systems for dataset 1 (D1) and dataset 2 (D2). P→ Precision, R→ Recall, F→ F˙score

tracking. Earlier, Mishra et al. (2013) measured translation annotation difficulty of a given sentence based on gaze input of translators used to label training data. The recent advancements in the literature discussed above, motivate us to explore gaze-based cognition for sentiment analysis.

We acknowledge that some of the well performing sentiment analyzers use Deep Learning techniques (like Convolutional Neural Network based approach by Maas et al. (2011) and Recursive Neural Network based approach by dos Santos and Gatti (2014)). In these, the features are automatically learned from the input text. Since our approach is feature based, we do not consider these approaches for our current experimentation. Taking inputs from gaze data and using them in a deep learning setting sounds intriguing, though, it is beyond the scope of this work.

## 3 Eye-tracking and Sentiment Analysis Datasets

We use two publicly available datasets for our experiments. Dataset 1 has been released by Mishra et al. (2016) which they use for the task of *sarcasm understandability* prediction. Dataset 2 has been used by Joshi et al. (2014) for the task of sentiment annotation complexity prediction. These datasets contain many instances with higher level nuances like presence of implicit sentiment, sarcasm and thwarting. We describe the datasets below.

### 3.1 Dataset 1

It contains 994 text snippets with 383 positive and 611 negative examples. Out of this, 350 are sarcastic or have other forms of irony. The snippets are a collection of reviews, normalized-tweets and quotes. Each snippet is annotated by **seven** participants with binary positive/negative polarity labels. Their eye-movement patterns are recorded with a high quality *SR-Research Eyelink-1000 eye-tracker* (sampling rate 500Hz). The annotation accuracy varies from **70%-90%** with a Fleiss kappa inter-rater agreement of **0.62**.

### 3.2 Dataset 2

This dataset consists of 1059 snippets comprising movie reviews and normalized tweets. Each snippet is annotated by **five** participants with positive, negative and objective labels. Eye-tracking is done using a low quality **Tobii T120** eye-tracker (sampling rate 120Hz). The annotation accuracy varies from **75%-85%** with a Fleiss kappa inter-rater agreement of **0.68**. We rule out the objective ones and consider 843 snippets out of which 443 are positive and 400 are negative.

### 3.3 Performance of Existing SA Systems Considering Dataset -1 and 2 as Test Data

It is essential to check whether our selected datasets really pose challenges to existing sentiment analyzers or not. For this, we implement two statistical classifiers and a rule based classifier to check the test accuracy of Dataset 1 and Dataset 2. The statistical classifiers are based on Support Vector Machine (SVM) and Näive Bayes (NB) implemented using Weka (Hall et al., 2009) and LibSVM (Chang and Lin, 2011) APIs. These are on trained on 10662 snippets comprising movie reviews and tweets, randomly collected from standard datasets released by Pang and Lee (2004) and Sentiment 140 (http://www.sentiment140.com/). The feature-set comprises traditional features for SA reported in a number of papers. They are discussed in section 4 under the category of *Sentiment Features*. The *in-house* rule based (RB) classifier decides the sentiment labels based on the counts of positive and negative words present in the snippet, computed using MPQA lexicon (Wilson et al., 2005). It also considers negators as explained by Jia et al. (2009) and intensifiers as explained by Dragut and Fellbaum (2014).

Table 1 presents the accuracy of the three systems. The F-scores are not very high for all the systems (especially for dataset 1 that contains more sarcastic/ironic texts), possibly indicating that the snippets in our dataset pose challenges for

existing sentiment analyzers. Hence, the selected datasets are ideal for our current experimentation that involves cognitive features.

## 4 Enhanced feature set for SA

Our feature-set into four categories *viz.* (1) Sentiment features (2) Sarcasm, Irony and Thwarting related Features (3) Cognitive features from eye-movement (4) Textual features related to reading difficulty. We describe our feature-set below.

### 4.1 Sentiment Features

We consider a series of textual features that have been extensively used in sentiment literature (Liu and Zhang, 2012). The features are described below. Each feature is represented by a unique abbreviated form, which are used in the subsequent discussions.

1. **Presence of Unigrams (NGRAM˙PCA) *i.e.*** Presence of unigrams appearing in each sentence that also appear in the vocabulary obtained from the training corpus. To avoid overfitting (since our training data size is less), we reduce the dimension to 500 using Principal Component Analysis.

2. **Subjective words (Positive˙words, Negative˙words) *i.e.*** Presence of positive and negative words computed against MPQA lexicon (Wilson et al., 2005), a popular lexicon used for sentiment analysis.

3. **Subjective scores (PosScore, NegScore) *i.e.*** Scores of positive subjectivity and negative subjectivity using SentiWordNet (Esuli and Sebastiani, 2006).

4. **Sentiment flip count (FLIP) *i.e.*** Number of times words polarity changes in the text. Word polarity is determined using MPQA lexicon.

5. **Part of Speech ratios (VERB, NOUN, ADJ, ADV) *i.e.*** Ratios (proportions) of verbs, nouns, adjectives and adverbs in the text. This is computed using NLTK[1].

6. **Count of Named Entities (NE) *i.e.*** Number of named entity mentions in the text. This is computed using NLTK.

---
[1] http://www.nltk.org/

7. **Discourse connectors (DC) *i.e.*** Number of discourse connectors in the text computed using an in-house list of discourse connectors (like *however*, *although* etc.)

### 4.2 Sarcasm, Irony and Thwarting related Features

To handle complex texts containing constructs irony, sarcasm and thwarted expectations as explained earlier, we consider the following features. The features are taken from Riloff et al. (2013), Ramteke et al. (2013) and Joshi et al. (2015).

1. **Implicit incongruity (IMPLICIT˙PCA) *i.e.*** Presence of positive phrases followed by negative situational phrase (computed using bootstrapping technique suggested by Riloff et al. (2013)). We consider the top 500 principal components of these phrases to reduce dimension, in order to avoid overfitting.

2. **Punctuation marks (PUNC) *i.e.*** Count of punctuation marks in the text.

3. **Largest pos/neg subsequence (LAR) *i.e.*** Length of the largest series of words with polarities unchanged. Word polarity is determined using MPQA lexicon.

4. **Lexical polarity (LP) *i.e.*** Sentence polarity found by supervised logistic regression using the dataset used by Joshi et al. (2015).

### 4.3 Cognitive features from eye-movement

Eye-movement patterns are characterized by two basic attributes: (1) Fixations, corresponding to a longer stay of gaze on a visual object (like characters, words *etc.* in text) (2) Saccades, corresponding to the transition of eyes between two fixations. Moreover, a saccade is called a *Regressive Saccade* or simply, *Regression* if it represents a phenomenon of going back to a pre-visited segment. A portion of a text is said to be *skipped* if it does not have any fixation. Figure 1 shows eye-movement behavior during annotation of the given sentence in dataset-1. The circles represent fixation and the line connecting the circles represent saccades. Our cognition driven features are derived from these basic eye-movement attributes. We divide our features in two sets as explained ahead.

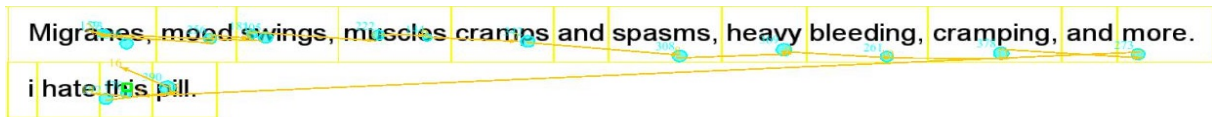

Figure 1: Snapshot of eye-movement behavior during annotation of an opinionated text. The circles represent fixations and lines connecting the circles represent saccades. Boxes represent Areas of Interest (AoI) which are words of the sentence in our case.

### 4.4 Basic gaze features

Readers' eye-movement behavior, characterized by fixations, forward saccades, skips and regressions, can be directly quantified by simple statistical aggregation (*i.e.,* computing features for individual participants and then averaging). Since these behaviors intuitively relate to the cognitive process of the readers (Rayner and Sereno, 1994), we consider simple statistical properties of these factors as features to our model. Some of these features have been reported by Mishra et al. (2016) for modeling sarcasm understandability of readers. However, as far as we know, these features are being introduced in NLP tasks like sentiment analysis for the first time.

1. **Average First-Fixation Duration per word (FDUR)** *i.e.* Sum of *first-fixation duration* divided by word count. First fixations are fixations occurring during the first pass reading. Intuitively, an increased first fixation duration is associated to more time spent on the words, which accounts for lexical complexity. This is motivated by Rayner and Duffy (1986).

2. **Average Fixation Count (FC)** *i.e.* Sum of fixation counts divided by word count. If the reader reads fast, the first fixation duration may not be high even if the lexical complexity is more. But the number of fixations may increase on the text. So, fixation count may help capture lexical complexity in such cases.

3. **Average Saccade Length (SL)** *i.e.* Sum of saccade lengths (measured by number of words) divided by word count. Intuitively, lengthy saccades represent the text being structurally/syntactically complex. This is also supported by von der Malsburg and Vasishth (2011).

4. **Regression Count (REG)** *i.e.* Total number of gaze regressions. Regressions correspond to both lexical and syntactic re-analysis (Malsburg et al., 2015). Intuitively,

regression count should be useful in capturing both syntactic and semantic difficulties.

5. **Skip count (SKIP)** *i.e.* Number of words skipped divided by total word count. Intuitively, higher skip count should correspond lesser semantic processing requirement (assuming that skipping is not done intentionally).

6. **Count of regressions from second half to first half of the sentence (RSF)** *i.e.* Number of regressions from second half of the sentence to the first half of the sentence (given the sentence is divided into two equal half of words). Constructs like sarcasm, irony often have phrases that are incongruous (*e.g. "The book is so great that it can be used as a paperweight"- the incongruous phrases are "book is so great" and "used as a paperweight"..* Intuitively, when a reader encounters such incongruous phrases, the second phrases often cause a surprisal resulting in a long regression to the first part of the text. Hence, this feature is considered.

7. **Largest Regression Position (LREG)** *i.e.* Ratio of the absolute position of the word from which a regression with the largest amplitude (in terms of number of characters) is observed, to the total word count of sentence. This is chosen under the assumption that regression with the maximum amplitude may occur from the portion of the text which causes maximum surprisal (in order to get more information about the portion causing maximum surprisal). The relative starting position of such portion, captured by LREG, may help distinguish between sentences with different linguistic subtleties.

### 4.5 Complex gaze features

We propose a graph structure constructed from the gaze data to derive more complex gaze features. We term the graph as *gaze-saliency graphs*.

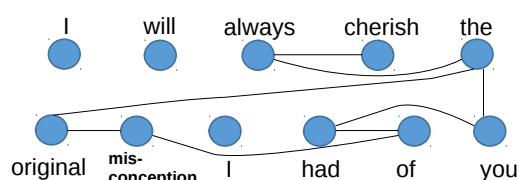

Figure 2: Saliency graph of a human annotator for the sentence *I will always cherish the original misconception I had of you.*

*A gaze-saliency graph for a sentence $S$ for a reader $R$, represented as $G = (V, E)$, is a graph with vertices $(V)$ and edges $(E)$ where each vertex $v \in V$ corresponds to a word in $S$ (may not be unique) and there exists an edge $e \in E$ between vertices $v_1$ and $v_2$ if $R$ performs at least one saccade between the words corresponding to $v1$ and $v2$. Figure 2 shows an example of such a graph.*

1. **Edge density of the saliency gaze graph (ED) i.e.** Ratio of number of edges in the gaze saliency graph and total number of possible links $((|V| \times |V| - 1|)/2)$ in the saliency graph. As, *Edge Density* of a saliency graph increases with the number of distinct saccades, it is supposed to increase if the text is semantically more difficult.

2. **Fixation Duration at Left/Source (F1H, F1S) i.e.** Largest weighted degree and second largest weighted degree of the saliency graph considering the fixation duration on the word of node $i$ of edge $E_{ij}$ as edge weight.

3. **Fixation Duration at Right/Target (F2H, F2S) i.e.** Largest weighted degree and second largest weighted degree of the saliency graph considering the fixation duration of the word of node $i$ of edge $E_{ij}$ as edge weight.

4. **Forward Saccade Word Count of Source (PSH, PSS) i.e.** Largest weighted degree and second largest weighted degree of the saliency graph considering the number of forward saccades between nodes $i$ and $j$ of an edge $E_{ij}$ as edge weight..

5. **Forward Saccade Word Count of Destination (PSDH, PSDS) i.e.** Largest weighted degree and second largest weighted degree of the saliency graph considering the total distance (word count) of forward saccades between nodes $i$ and $j$ of an edge $E_{ij}$ as edge weight.

6. **Regressive Saccade Word Count of Source (RSH, RSS) i.e.** Largest weighted degree and second largest weighted degree of the saliency graph considering the number of regressive saccades between nodes $i$ and $j$ of an edge $E_{ij}$ as edge weight.

7. **Regressive Saccade Word Count of Destination (RSDH, RSDS) i.e.** Largest weighted degree and second largest weighted degree of the saliency graph considering the number of regressive saccades between nodes $i$ and $j$ of an edge $E_{ij}$ as edge weight.

The "highest and second highest degree" based gaze features derived from saliency graphs are motivated by our qualitative observations from the gaze data. Intuitively, the highest weighted degree of a graph is expected to be higher if some phrases have complex semantic relationships with others.

## 4.6 Features Related to Reading Difficulty

Eye-movement during reading text with sentiment related nuances (like sarcasm) can be similar to text with other forms of difficulties. To address the effect of sentence length, word length and syllable count that affect reading behavior, we consider the following features.

1. **Readability Ease (RED) i.e.** Flesch Readability Ease score of the text (Kincaid et al., 1975). Higher the score, easier is the text to comprehend.

2. **Sentence Length (LEN) i.e.** Number of words in the sentence.

We now explain our experimental setup and results.

## 5 Experiments and results

We test the effectiveness of the enhanced feature-set by implementing three classifiers *viz.,* SVM (with linear kernel), NB and Multi-layered Neural Network. These systems are implemented using the Weka (Hall et al., 2009) and LibSVM (Chang and Lin, 2011) APIs. Several classifier hyperparameters are kept to the default values given in Weka. We separately perform a 10-fold cross validation on both Dataset 1 and 2 using different sets of feature combinations. The average F-scores for the class-frequency based random classifier are **33%** and **46.93%** for dataset 1 and dataset 2 respectively.

| Classifier | Näive Bayes | | | SVM | | | Multi-layer NN | | |
|---|---|---|---|---|---|---|---|---|---|
| | **Dataset 1** | | | | | | | | |
| | **P** | **R** | **F** | **P** | **R** | **F** | **P** | **R** | **F** |
| Uni | 58.5 | 57.3 | 57.9 | 67.8 | 68.5 | 68.14 | 65.4 | 65.3 | 65.34 |
| Sn | 58.7 | 57.4 | 58.0 | 69.6 | 70.2 | 69.8 | 67.5 | 67.4 | 67.5 |
| Sn + Sr | 63.0 | 59.4 | 61.14 | 72.8 | 73.2 | 72.9 | 69.0 | 69.2 | 69.1 |
| Gz | 61.8 | 58.4 | 60.05 | 54.3 | 52.6 | 53.4 | 59.1 | 60.8 | 60 |
| Sn+Gz | 60.2 | 58.8 | 59.2 | 69.5 | 70.1 | 69.6 | 70.3 | 70.5 | 70.4 |
| **Sn+ Sr+Gz** | **63.4** | **59.6** | **61.4** | **73.3** | **73.6** | **73.5** | **70.5** | **70.7** | **70.6** |
| | **Dataset 2** | | | | | | | | |
| Uni | **51.2** | **50.3** | **50.74** | 57.8 | 57.9 | 57.8 | 53.8 | 53.9 | 53.8 |
| Sn | 51.1 | 50.3 | 50.7 | 62.5 | 62.5 | 62.5 | 58.0 | 58.1 | 58.0 |
| Sn+Sr | 50.7 | 50.1 | 50.39 | 70.3 | 70.3 | 70.3 | 66.8 | 66.8 | 66.8 |
| Gz | 49.9 | 50.9 | 50.39 | 48.9 | 48.9 | 48.9 | 53.6 | 54.0 | 53.3 |
| Sn+Gz | 49.9 | 50.9 | 48.5 | 48.9 | 48.9 | 48.9 | 53.6 | 54.0 | 53.8 |
| **Sn+ Sr+Gz** | 50.2 | 49.7 | 50 | **71.9** | **71.8** | **71.8** | **69.1** | **69.2** | **69.1** |

Table 2: Results for different feature combinations. (P,R,F)→ Precision, Recall, F-score. Feature labels Uni→Unigram features, Sn→Sentiment features, Sr→Sarcasm features and Gz→Gaze features along with features related to reading difficulty

The classification accuracy is reported in Table 2. We observe the maximum accuracy with the complete feature-set comprising Sentiment, Sarcasm and Thwarting, and Cognitive features derived from gaze data. For this combination, SVM outperforms the other classifiers. **The novelty of our feature design lies in (a) First augmenting sarcasm and thwarting based features (*Sr*) with sentiment features (*Sn*), which shoots up the accuracy by 3.1% for Dataset1 and 7.8% for Dataset2 (b) Augmenting gaze features with *Sn+Sr*, which further increases the accuracy by 0.6% and 1.5% for Dataset 1 and 2 respectively, amounting to an overall improvement of 3.7% and 9.3% respectively**.

Since the best and the second best features are close in terms of accuracy for dataset 1 (difference of 0.5%), we perform a statistical significance test using **McNemar test** ($\alpha = 0.05$). The difference in the F-scores turns out to be significant with $p = 0.0001$.

We also perform a *chi-squared test* based feature significance analysis, shown in Table 3. For dataset 1, **10 out of the top 20** ranked features are gaze-based features and for dataset 2, **7 out of top 20** features are gaze-based, as shown in bold letters.

| Rank | Dataset 1 | Dataset 2 |
|---|---|---|
| 1 | PosScore | LP |
| 2 | LP | Negative_Words |
| 3 | NGRAM_PCA_1 | Positive_Words |
| 4 | **FDUR** | NegCount |
| 5 | **F1H** | PosCount |
| 6 | **F2H** | NGRAM_PCA_1 |
| 7 | NGRAM_PCA_2 | IMPLICIT_PCA_1 |
| 8 | **F1S** | FC |
| 9 | ADJ | **FDUR** |
| 10 | **F2S** | NGRAM_PCA_2 |
| 11 | NGRAM_PCA_3 | **SL** |
| 12 | NGRAM_PCA_4 | **LREG** |
| 13 | **RSS** | **SKIP** |
| 14 | **PSDH** | **RSF** |
| 15 | **PSDS** | **F1H** |
| 16 | IMPLICIT_PCA_1 | RED |
| 17 | **LREG** | LEN |
| 18 | **SKIP** | PUNC |
| 19 | IMPLICIT_PCA_2 | IMPLICIT_PCA_2 |

Table 3: Features as per their ranking for both Dataset 1 and Dataset 2. Integer values $N$ in NGRAM_PCA_N and IMPLICIT_PCA_N represent the $N^{th}$ principal component.

### 5.1 Importance of cognitive features

To study whether the cognitive features actually help in classifying complex output as hypothesized earlier, we repeat the experiment on a held-out dataset, randomly derived from Dataset-1. It has 294 text snippets out of which 131 contain complex constructs like irony/sarcasm and rest of the snippets are relatively simpler. We choose

|        | Irony | Non-Irony |
|--------|-------|-----------|
| Sn     | 58.2  | 75.5      |
| Sn+Sr  | 60.1  | 75.9      |
| Gz+Sn+Sr | 64.3 | 77.6     |

Table 4: F-scores on held-out dataset for Complex Constructs (Irony), Simple Constructs (Non-irony)

SVM, our best performing classifier, with similar configuration as explained in section 5. As seen in Table 4, the relative improvement of F-score, when gaze features are included, is **6.1%** for complex texts and is **2.1%** for simple texts (all the values are statistically significant with $p < 0.05$ for McNemar test, except $Sn$ and $Sn + Sr$ for Non-irony case.). This demonstrates the efficacy of the gaze based features.

## 6    Error Analysis

Errors committed by our system arise from multiple factors starting from limitations of the eye-tracker hardware to errors committed by linguistic tools and resources. Moreover, aggregating various eye-tracking parameters to extract the cognitive features may have caused loss of information. For example, the graph based features are computed for each participant and eventually averaged to get the graph features for a sentence, thereby not leveraging the power of individual eye-movement patterns.

## 7    Feasibility of our approach

Since our method requires gaze data from human readers to be available, the methods practicability becomes questionable. We present our views on this below.

### 7.1    Availability of Mobile Eye-trackers

Availability of inexpensive embedded eye-trackers on hand-held devices has come close to reality now. This opens avenues to get eye-tracking data from inexpensive mobile devices from a huge population of online readers non-intrusively, and derive cognitive features to be used in predictive frameworks like ours. For instance, *Cogisen: (http://www.sencogi.com)* has a patent (ID: EP2833308-A1) on "eye-tracking using inexpensive mobile web-cams".

### 7.2    Applicability Scenario

We believe, mobile eye-tracking modules could be a part of mobile applications built for e-commerce, online learning, gaming *etc.* where automatic analysis of online reviews calls for better solutions to detect and handle linguistic nuances in sentiment analysis setting. To give an example, let's say a book gets different reviews on Amazon. Our system could watch how readers read the review using mobile eye-trackers, and thereby, decide the polarity of opinion, especially when sentiment is not expressed explicitly (*e.g.,* using strong polar words) in the text. Such an application can horizontally scale across the web, helping to improve automatic classification of online reviews.

### 7.3    Getting Users' Consent for Eye-tracking

Eye-tracking technology has already been utilized by leading mobile technology developers (like Samsung) to facilitate richer user experiences through services like *Smart-scroll* (where a user's eye movement determines whether a page has to be scrolled or not) and *Smart-lock* (where user's gaze position decided whether to lock the screen or not). The growing interest of users in using such services takes us to a promising situation where getting users' consent to record eye-movement patterns will not be difficult, though it is yet not the current state of affairs.

## 8    Conclusion

We combined traditional sentiment features with (a) different textual features used for sarcasm and thwarting detection, and (b) cognitive features derived from readers' eye movement behavior. The combined feature set improves the overall accuracy over the traditional feature set based SA by a margin of 3.6% and 9.3% respectively for Datasets 1 and 2. It is significantly effective for text with complex constructs, leading to an improvement of 6.1% on our held-out data. In future, we propose to explore (a) devising deeper gaze-based features and (b) *multi-view* classification using independent learning from linguistics and cognitive data. We also plan to explore deeper graph and gaze features, and models to learn complex gaze feature representation. Our general approach may be useful in other problems like emotion analysis, text summarization and question answering, where textual clues alone do not prove to be sufficient.

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
