# Peer review of "Leveraging Cognitive Features for Sentiment Analysis"

_CoNLL 2016 — decision unknown_

[Official Review · Reviewer 1 · rating 4 · confidence 3]
soundness 4 · originality 4 · clarity 4 · impact 3 · substance 4 · appropriateness 5 · meaningful comparison 4 · replicability 2 · presentation format Poster

This paper proposed a very interesting idea of using cognitive features for
sentiment analysis and sarcasm detection. More specifically, the eye-movement
patterns of human annotators are recorded to derive a new set of features. The
authors claim that this is the first work to include cognitive features into
the NLP community. 

Strength: 
1. The paper is generally well written and easy to follow
2. Very interesting idea which may inspire research in other NLP tasks.

Weakness:
1. The motivation of using cognitive features for sentiment analysis is not
very well justified. I can imagine these features may help reflect the reading
ease, but I don't see why they are helpful in detecting sentiment polarities.
2. The improvement is marginal after considering cognitive features by
comparing Sn+Sr+Gz with Sn+Sr.
3. Although the authors discussed about the feasibility of the approach in
Section 7, but I'm not convinced, especially about the example given in section
7.2, I don't see why this technique is helpful in such a scenario.

[Official Review · Reviewer 2 · rating 4 · confidence 2]
soundness 5 · originality 4 · clarity 5 · impact 4 · substance 4 · appropriateness 5 · meaningful comparison 5 · replicability 4 · presentation format Oral Presentation

This paper is about introducing eye-tracking features for sentiment analysis as
a type of cognitive feature.  I think that the idea of introducing eye-tracking
features as a proxy for cognitive load for sentiment analysis is an interesting
one.  

I think the discussion on the features and comparison of feature sets is clear
and very helpful.  I also like that the feasibility of the approach is
addressed in section 7.

I wonder if it would help the evaluation if the datasets didn't conflate
different domains, e.g., the movie review corpus and the tweet corpus.             
For one
it might improve the prediction of movie review (resp. tweets) if the tweets
(resp. movie reviews) weren't in the training.              It would also make the
results
easier to interpret.  The results in Table 2 would seem rather low compared to
state-of-the art results for the Pang and Lee data, but look much better if
compared to results for Twitter data.

In Section 3.3, there are no overlapping snippets in the training data and
testing data of datasets 1 and 2, right?  Even if they come from the same
sources (e.g., Pang & Lee and Sentiment 140).

Minor: some of the extra use of bold is distracting (or maybe it's just me);